# Changes in the Rheological and Adhesive Properties of Epoxy Resin Used in the Technology of Reinforcement of Structural Elements with CFRP Tapes

**DOI:** 10.3390/ma16237408

**Published:** 2023-11-28

**Authors:** Andrzej Szewczak

**Affiliations:** Faculty of Civil Engineering and Architecture, Lublin University of Technology, Nadbystrzycka Str. 40, 20-618 Lublin, Poland; a.szewczak@pollub.pl; Tel.: +48-815-384-428

**Keywords:** profilometry, epoxy resins, sonication, microsilica, carbon nanotubes, adhesion

## Abstract

Nowadays, FRP composites are widely applied in the construction industry. Their inherent characteristics are used to strengthen structural elements made of various materials and to increase their durability. The following paper contains the results obtained in a conducted research program aimed at determining the ability to improve the durability of epoxy resin modified with the sonication process, microsilica and carbon nanotubes. The adhesive modified was used to adhere a CFRP tape to a concrete surface. Changes in the viscosity, physicochemical parameters and adhesion of the resin were studied. Selected parameters of the concrete substrate prepared using the sandblasting method, determined with the contact profilometry, were also taken into account. During the tests, attention was paid to the thorough execution and preparation of the samples. As a result of the research, it was demonstrated that the adhesion of the modified epoxy adhesive to concrete could be increased by approximately 28% in the case of the addition of carbon nanotubes and by up to 66% in the case of the addition of microsilica. The modifications used, in addition to increasing the adhesion of the resin to the concrete substrate, were also aimed at reducing the weakening of the adhesive joints caused by oxidation of the resin over time. The results obtained will serve as a basis for evaluating the possibility of their use in the practical reinforcement of structural reinforced-concrete elements.

## 1. Introduction

FRP (Fiber-Reinforced Polymer) composites are a specific and very interesting group of composite materials. They are produced in the form of materials combining the properties of selected fibers (carbon, glass, aramid, basalt, etc.) as a filling and as matrixes in the form of resins, usually epoxy, phenolics and polyester. Their very good mechanical parameters, primarily their high tensile strength and modulus of elasticity (Table 1), are the result of using fibers [1,2,3,4,5]. They are also characterized by a low weight-to-strength ratio. They allow for the reinforcement of structural elements made of various materials, such as steel, wood or concrete. At the same time, they are characterized by a low longitudinal strain of 0.5–2.5%, which is due to the presence of a polymer binder in which the fibers are embedded (Figure 1) [2,5,6,7].

The relatively small value of deformation influences the specific manner of their failure under testing or working conditions in the structure. FRP composites are most often found in the form of tapes, mats or rods (Figure 2), which, when properly adhered to structural elements, can significantly increase their load-bearing capacity, especially in terms of tension, and durability. Numerous studies on such reinforcements are currently being conducted [1,2,8,9]. They largely focus on testing elements strengthened or reinforced with FRP composites subjected to loading and analyzing the resulting deformations and stresses. The important conclusions related to the research on CFRP composites (which were also the subject of research described later in the article) are their greater resistance to fatigue loads, lower susceptibility to creep and overall greater durability compared to other FRP composites used in construction [10,11].

In building structures, FRPs are used in several distinct roles:−Reinforcing the structure by gluing and/or anchoring tapes and mats on the surface of the elements [12];−Reinforcement of reinforced-concrete elements with FRP bars [13,14];−Prestressing of tapes and bars [13,14,15];−Prestressed structures with reinforcement in the form of prestressed cables together with the use of appropriate anchoring systems [16,17,18].

The bonding of composites to reinforced components is achieved in several ways. The most popular of these is bonding using adhesives polymers, such as epoxy resins [2,4,6]. In this case, the load-bearing capacity and durability of such a joint depends on both the adhesion of the resin itself and the roughness of the substrate. It is also possible to strengthen the bond through active or passive anchoring of the tape to, e.g., (Figure 3) a reinforced concrete beam. In this case, the load-bearing capacity of the tape restrained in the holder is influenced by the tension force of the tape and the load-bearing capacity of the anchors fixing the holder in the element [13,14,15]. Since adhesion (lat. *adhesio*) is a difficult phenomenon to define unambiguously, different models of it are analyzed, depending on individual elemental factors. The works in [19,20,21] define individual definitions of adhesion (Figure 4). However, a general definition, due to the complex nature of this phenomenon, is not possible. Depending on the preparation of the surface for bonding, the type of adhesive and its chemical composition, it is possible to increase or decrease the contribution of individual types of adhesion to the overall adhesion to the substrate.

In the bonding between concrete and CFRP tape, mechanical adhesion and adsorption play the greatest role. This is primarily due to the topography of the substrate and the physicochemical state of the adhesive at the time of application and after curing. Changes in mechanical adhesion are possible due to modifications of the surface to which FRP composites are adhered. Variations in roughness lead to a system in which the glue and substrate can dovetail each other [23,24]. The definition and model of mechanical adhesion were first described by McBain and Hopkins in 1925 [25,26]. Roughness is understood as the differential state of irregularity on the material surface and can be described using height and amplitude parameters [27]. There are many works describing the surface condition and the effects of various modifications to improve mechanical adhesion. Surface preparation may lead to changes in the roughness parameters and an increase in mechanical adhesion. In concrete surfaces, the most commonly used methods of mechanical surface treatment include [23,24] cleaning, bulk abrasives, abrasive blasting, shot peening, brushing, grinding and milling.

The effectiveness of bonding FRP composites to concrete also depends largely on the occurrence of adsorption forces. According to Figure 4, they mainly depend on the presence of chemical bonds between atoms contained in the solid, liquid or gas phase [19,20]. Concrete surface treatment methods not only lead to the development of a rough surface, but also stimulate its energy state. The relocation of electrons in the surface layer causes their polarization and the formation of centers that are more easily involved in chemical reactions with atoms located in the polymer. This is obviously due to the distribution of electrons that can be exchanged between the adhesive and the substrate. Important in the description of this phenomenon are the assumptions of Lewis’ theory of acids and bases [20]. The theory assumes the exchange of electrons between the concrete substrate (donor) and the adhesive (electron acceptor). The intensity of this process depends on the ability of individual layers to exchange charges. In this way, it is important not only to properly prepare the surface but also that the adhesive has the ability to bond to the substrate. [23,28,29].

Epoxy resins have functional groups at the ends of polymer chains. Their structure also contains free electrons in the form of clouds located between polymer chains. When applying a resin to a substrate, they may participate in the formation of bonds. It is possible to appropriately modify the polymer to achieve this effect. One way is to use sonication, as the author also described in [30]. Cavitation leads to several phenomena. The polymer chains are straightened and broken. Their functional groups and free electrons from the interior structure are relocated. After sonication, the polymer structure returns to its stable internal structure relatively quickly. However, observations show that the structure is more ordered and the glue itself becomes more reactive. The release of electrons enables their participation in the formation of permanent adsorption bonds, e.g., covalent bonds and those related to van der Waals forces. These processes occur at the interfaces of the liquid, solid and gas phases. These phenomena are relatively difficult to explain, which is also related to the complex nature of adhesion itself [23].

The process of changing the reactivity of the adhesive can also be influenced by so-called powder fillers, which are additives to polymers that change their basic properties, primarily functional, rheological and mechanical [31,32,33]. Depending on the amount of filler, it is possible to further reduce to some extent the amount of polymer used in processing. Among the most popular powder fillers today are ground limestone, dolomite, chalk, clay derived from ground brick and other ceramic materials, microsilica, quartz powder, granite powder, basalt powder, gypsum, mica or soot [32,33,34,35]. The amount of filler is most often determined experimentally. Most often, it is possible to determine the upper and lower limits for the use of filler, the percentage of which depends on the weight of the polymer itself. It is also important to determine which properties of the polymer a given filler should favorably affect. Modification of epoxy resin-based adhesive polymers with the addition of fillers should result in an increase in the mechanical parameters of the adhesive and its adhesion to the target substrate, such as steel or concrete. An important issue is to determine the operating conditions of such a joint after it is made. Filler molecules, as a rule, form permanent or temporary chemical bonds with molecules in the polymer chain [32,36]. The movement of electrons coming from filler molecules determines the behavior of polymer functional groups during polymer crosslinking. In some cases, the addition of a filler has the effect of prolonging the aging effect of the polymer in the joint, which is one of the frequently analyzed problems related to the use of polymers in many industrial fields. This phenomenon is primarily associated with the weakening of chemical bonds and a loss of polymer properties under the influence of environmental factors, i.e., varying temperature or UV radiation [34,36,37]. Other physical parameters of the fillers, i.e., specific surface area, shape or density, allow increased adhesion of the adhesive to the target surface to be achieved. With the help of fillers, it is possible to change both the mechanical adhesion of the adhesive to the substrate and the adhesion resulting from the occurrence of various bonds and chemical interactions.

Proper mixing of the filler with the polymer can be achieved in several ways. Among these methods are mechanical mixing, using pressure changes, or sonication. As a result of mixing the phases, i.e., a polymer (liquid) and a filler (solid), a homogeneous mixture is formed, which is then subjected to a crosslinking reaction involving the hardener [30,36,38,39].

The aim of the research presented in this article was to analyze the possibility of increasing the adhesion of the selected epoxy glue to the sandblasted concrete surface. This effect was achieved by modifying the adhesive through the addition of microsilica, multilayer carbon nanotubes and the process of sonication. The reason for choosing such fillers is their different structure. Microsilica is characterized by spherical particles with a polar structure, while nanotubes have a crosslinked structure that can be broken down and straightened. In this way, the expected effect of the interaction between the fillers and the polymer was different. Glue with a base addition of quartz flour was selected for the study for use in the system of strengthening of structural elements with CFRP tapes. An indicator of the effectiveness of bonding with the modified adhesive was the peel strength of CFRP tape fragments bonded to concrete, tested using the pull-off method. Possible changes occurring in the resin structure under the influence of the described modifications were determined. Changes in rheological and physicochemical parameters were examined as auxiliary parameters determining the bonding efficiency: viscosity, contact angle, temperature and surface free energy. The results also depended on surface roughness parameters determined via contact profilometry.

## 2. Materials and Methods

### 2.1. Materials Used and Mixtures

The main material that was tested was an epoxy adhesive (SikaDur, Warsaw, Poland) used in a system for reinforcing structural members with CFRP tapes (designated as the SD in the study). This adhesive is characterized by the presence of quartz powder as a base filler. The binder is an epoxy resin. The amount of powder in the total volume of the adhesive is about 60%. The properties of the adhesive are shown in Table 2. Curing was carried out with the participation of a hardener in the form of a gray mass, which is trimetylohexa—1.6—diamine, in the amount of 33%.

In the initial phase of the study, the adhesive was subjected to the three modifications:Modification by means of sonication;Modification with the addition of microsilica in the amount of 0.5% in relation to the mass of resin;Modification with the addition of carbon nanotubes in the amount of 0.1% in relation to the mass of resin.

In series with filler additives, sonication was used as a method to allow the adhesive to mix with the fillers. Similar studies were described in [30]. The parameters of the fillers used in the study are as follows:Microsilica, BASF (BASF, Ludwigshafen, Germany), with a density of 2.2 g/cm^3^, an average particle diameter of 0.1µm and a specific surface area of 20,000 m^2^/kg.NanocylTM NC7000 carbon nanotubes, manufacturer NANOCYL (NANOCYL, Sambreville, Belgium), with a density of 1.3–1.4 g/cm^3^, average diameter of 9.5 nm, average length of 1.5 μm and specific surface area of 250–300 m^2^/g.

The formulations and designations of each series of samples are shown in Table 3.

### 2.2. Methodology

The research program included several successive research phases:Preparation of adhesive in the process of sonication—sonication was carried out using a portable sonicator UP 400S (Hielscher Ultrasonics Gmbh, Teltow, Germany) with a power of 400 W, amplitude equal to 1, range of used power 100% and frequency 20 kHz; in the case of the modified series the sonication time was 6 min, which allowed a uniform structure composed of mixed substances to be obtained;Measurement of adhesive viscosity—this is a parameter largely determining the adhesion of the adhesive to the substrate during its application, and it was determined for all series; the value measured for the SD series at 22 °C served as a reference value, and was measured using a stationary rotational viscosity meter type H2 from FungiLab (FungiLab, Barcelona, Spain). Additionally, from the moment the sonicator was turned off, viscosity and temperature were measured (the PT-105 laboratory thermometer from Elmetron, Zabrze, Poland) at 5 min intervals until the adhesive cooled to the initial temperature of 22 °C; a diagram showing the stand for sonication and viscosity measurements is shown in Figure 5.Measurement of surface free energy—the SFE was used to describe in detail the phenomena that may occur at the interface between the adhesive and the concrete substrate during the application. From among several methods of SFE measurement described in the literature, the Owens–Wendt method was chosen. Thanks to this, it is possible to determine the polar and dispersion components at the phase interface on the basis of measurements of the wetting angles of drops of two reference liquids with known dispersive–polar parameters; in the described studies, these were distilled water and diiodomethane (Figure 6 and Figure 7); these parameters are non-negligible in explaining the phenomena related to adsorption, which, next to mechanical adhesion, is most responsible for the adhesion of an epoxy adhesive to different types of substrates. For the measurements, 5 samples of 10 cm diameter and 1 cm thickness were made of each resin; then, on each disk, 5 wetting angle measurements were made for each liquid using the PGX goniometer (Klima-test, Wroclaw, Poland) with a measurement accuracy of ±1°.Measuring the adhesion of the adhesive to the concrete substrate—after taking post-measurements before the hardening, the next step was to check the effectiveness of the adhesive with regard to the possibility of increasing the adhesion of the resin on the sandblasted concrete surface. In order to make the tests possible, the samples were made of concrete of class C30/37; the concrete class was dictated by the adhesive manufacturer’s requirement that the concrete peel strength should be at least 2 MPa. In the case of concretes of lower classes, this condition would not always be met. The concrete samples were dried and subjected to the sandblasting process 200 days after being formed; for the surface prepared in this way, the profilometric measurements were taken using a T8000 RC120-400 contact profilometer from Homme-Etamic (Charlotte, NC, USA), then 2.5 × 3 cm pieces of the CFRP tape were glued to the concrete surface using modified adhesives. The size of the CFRP tape specimens was dictated by the parameters of the pull-off Dynatest device (Gainesville, FL, USA) having a load range of 0–25 kN, which was used to detach them from the concrete surface. Four detachment tests were performed for each series; the device and the view of the tape on the concrete surface are shown in Figure 8.

## 3. Results and Discussion

### 3.1. Rheological and Adsorption Properties

The results of viscosity measurements at a reference temperature of 22 °C are shown in Figure 9, while Figure 10 shows the course of viscosity and temperature changes over time.

From the data presented in Figure 9, it is clear that the applied modifications noticeably and significantly changed the viscosity. However, it is worth noting that these changes varied in scope and course. A number of processes and phenomena occurring during sonication are responsible for the viscosity changes. However, depending on the formulation of the modified adhesive, the mechanisms involved in cavitation are different. Of great importance in this case is the presence of quartz powder in the epoxy resin. The effect of its occurrence is a specific effect of ultrasonic energy on the structure of the polymer chains contained in the resin. In contrast to the phenomena occurring in pure resin, described, among others, in [30,40], the quartz powder disturbs the process of intrinsic ultrasounds propagation. Part of the energy actually causes partial disruption of the chains and their reconnection, mainly due to the formation of gas bubbles containing oxygen and molecules of simple hydrocarbons included in the single mers. The free electrons released during this time contribute to faster and easier joining of the broken chains into new, active structures, which can later more easily bond to the concrete substrate. Observations made during sonication of the adhesive prepared according to the SD/S formulation show that part of the energy coming from the ultrasound tip is absorbed by spherical quartz powder particles, as also described in [41,42]. Its semicrystalline structure supports the transfer of energy in the form of vibrations between the following particles. In this way, ultrasonic waves can propagate in two ways, at different speeds, taking into account the density of the resin and quartz powder, as two media with different phases (liquid and solid) [30,34,43]. It is worth noting that all these processes are very dynamic, while their final effect is visible after stabilization of the adhesive structure and during curing. In the case of the SD/S series, a decrease in viscosity of about 25% was noted. Due to the lack of filler additives, such a change is due to a more efficient organization of the mers, a reduction in the distance between them, and fewer voids in their three-dimensional structure. As a result, it may be smoother for the adhesive to penetrate the irregularities of substrates with more varied topography. Nevertheless, as studies [44,45,46] indicate, this effect may depend on the duration of sonication. Microsilica and carbon nanotubes affect this mechanism differently. This is shown in the graph in Figure 9 and Figure 10. The presence of microsilica, which is a form of SiO_2_ (similar to quartz powder), allows better filling of the free spaces between the epoxy adhesive chains that remain after the sonication process. In addition, microsilica particles are able to form temporary van der Waals bonds with the mers, which, in effect, due to the thickening of the resin structure, contributes to an increase in viscosity of about 62%. In the studies reported in [47,48], the exact course of possible reactions occurring between the microsilica and the epoxy resin was determined. Such a change may prove to be efficient in the case of bonding on a relatively smooth surface or with a regular distribution of hills and depressions. Such a nature of topography characterizes, for example, a sanded concrete surface. The adhesive is able to stay on the surface longer without the phenomenon of excessive spreading. Carbon nanotubes also contribute to an increase in viscosity, and more than threefold. This is the result of two phenomena. The first concerns, as also described in [30,41,49,50,51], the effect of ultrasounds on the carbon nanotubes themselves. Their multilayer structure is broken down in this process. Occurring regularly, the double chemical bonds between carbon atoms are disconnected [48,50]. In this way, this filler significantly increases its surface area, forming a kind of network that is able to interpenetrate the network of polymer chains. In the studies described in [52,53], it was shown based on FTIR analyses that carbon nanotubes do not interfere with the course of the hardening reaction of epoxy resins. In the case of epoxies, barriers may occur at the interfaces of the liquid resin and carbon nanotube phases, which can be overcome at the higher temperatures at which the nanotubes are introduced into the resin. Issues related to changes in temperature and viscosity during sonication, described in [54], among others, speak of the dynamic nature of the changes. Over the course of sonication, the viscosity may change its value (decrease and increase) many times at the beginning, until a certain steady trend of decrease (while the temperature increases) is stabilized. Such mechanisms of filler distribution in the polymer structure also affect the heat release rate of adhesives made according to particular formulations (Figure 10). In the case of the SD/S and SD/S/N series, the time for a given adhesive to reach a temperature of 22 °C is the same, despite some differences between viscosity values. The SD/S/M series showed a significantly longer heat release time. This mechanism was influenced by a structure composed of a polymer, microsilica and quartz powder. It is also worth noting that the series with added fillers showed an overall lower temperature when the sonicator was turned off. This may reflect the significant absorption of some of the heat released by the sonicator by microsilica particles and carbon nanotubes [51]. Thus, there is no danger of overheating the adhesive, e.g., above the ignition temperature of the epoxy resin.

Further analysis of the adsorption properties of the adhesives of the various formulations included examinations of wetting angles and surface free energy. The results presented in Figure 11 provide some more accurate insight into how adhesives can penetrate the irregularities of the concrete substrate.

Figure 11 brings some interesting conclusions. First of all, it is important to note the values of the individual SFE components. A characteristic feature of SD adhesives is the definite predominance of the value of the dispersive component over the polar component, also with regard to the overall SFE value, which, as is known, is the sum of these two components. This is a characteristic feature of adhesives with quartz powder additives, where the way in which the adhesive penetrates into the irregularities of the substrate is mainly determined by the way in which the adhesive coating adapts to the topography of the concrete. This feature is related precisely to the dispersion of the adhesive. The epoxy resin bonding to the substrate is responsible for the possibility of creating adhesion bonds resulting from chemical bonds (permanent and temporary) in this case [55,56]. A similar mechanism is presented by the resin in the fabrication of FRP composites and fiber wrapping, as demonstrated by studies described in [57]. However, the different series showed some different relationships in this regard. As can be seen in the presented graph, all modified series showed a non-significant decrease in SFE values in the range of 7.2–8.2%. They are relatively low in SD adhesive, hence the low values of the polar component. The values of the polar component changed by 23.2% for the SD/S series, 45.6% for the SD/S/M series and 11.7% for the SD/S/N series, respectively. In all cases, the observed relationships are due to the sonication process. According to previous conclusions, during this process, the polymer structure undergoes secondary reorganization and a reduction in the distances between chains. Some of the electrons that were free at the time of “polymer rest” find their place on the orbitals of atoms belonging to the epoxy mass structure [46]. This results in them having less participation in the exchange of electrons with atoms on the concrete surface. In the SD/S/M and SD/S/N series, in addition, part of the electrons also enters in their exchange processes with filler molecules. As a result, the lower values of the polar component suggest a smaller share of the adsorption. However, during the bonding itself, it is possible to reorganize the arrangement of polymers in the adhesive at the adhesive curing stage, so the actual contribution of the adsorption was possible after testing the samples in the pull-off test. As for the dispersion component, whose changes were less varied (5.3%, 2.8% and 6.8%, respectively), this effect was due to the behavior of the resin binder, which is, in a way, a kind of carrier of quartz powder grains. The lower activity of the adhesive and its more dense formulation also favor lower dynamics of adhesive dispersion on the target substrate. However, due to the fact of the high complexity of the dispersive–polar phenomena, not all phenomena that are possible to determine from the analysis of SFE results necessarily affect the deterioration in the glue adhesion.

The analysis of the results of wetting angles (Figure 12) should be referred to the nature of the measurement liquids used. Distilled water is considered a polar liquid [58]. Its polar component of SFE is 51 mJ/m^2^, with a total SFE of 72.8 mJ/m^2^. The opposite is true for diiodomethane, which, as a highly dispersive liquid, has a total SFE value of 50.8 mJ/m^2^, of which the dispersive component is as high as 48.5 mJ/m^2^ [23,58,59]. Therefore, the angle values were definitely different. Nevertheless, they made it possible to clarify the nature of the overall values of SFE and its components presented in Figure 12. If bonding is analyzed as a process involving the penetration, covering and filling of irregularities in the concrete substrate by the adhesive mass, lower values of wetting angles are indicated. However, sometimes this may not be a sufficient conclusion. Increasing the value of the angle results in a decrease in the value of the SFE, and therefore also in the work (energy) that needs to be carried out so that an adhesive layer can be formed on a unit area of a substrate. In this case, the combination of dispersive and polar properties influences the adhesive performance and the way the adhesive covers the substrate irregularities. The modified series showed the increase of wetting angle values for both distilled water (6.5% for the SD/S series, 11.8% for the SD/S/M series and 2.6% for the SD/S/N series) and for diiodomethane. In this case, the changes were 8.5%, 8.2% and 29.2%, respectively. These changes are not significant, except for the D angle for the SD/S/N series, so they do not change the adhesive behavior by significantly worsening its adhesive performance. It is also important to still remember the decisive contribution of mechanical adhesion when using the SD adhesive. Changes in the SFE values, which relate to the behavior of the adhesive on the concrete surface, also affect the phenomena occurring at the interface between the epoxy mass and a quartz powder grain. Lower values of SFE result in an easier coating of these grains with the polymer, also filling the free spaces between grains. These processes most likely still occur at the stage of glue cooling after the cessation of sonication. Very similar dependencies also characterize other materials with significant viscosity, which are applied to, e.g., fibers or aggregate grains [57,59]. The demonstrated viscosity changes are therefore also the result of these phenomena. At this stage of the analysis, one of the situations shown in Figure 10 is explained. The SD/S/M adhesive showed a significantly longer temperature recovery period to 22 °C. At the same time, it was characterized by the lowest value of the polar component of SFE. On this basis, a conclusion can be derived suggesting a longer and more dynamic formation of the structure of the modified adhesive, which is formed due to the organization of particles of a polymer, microsilica and quartz powder. The prolonged cooling time covers the temperature range from 22 to 27 °C, so there is a real possibility of chemical reactions leading to the creation of temporary chemical bonds. These reactions are likely to be partially exothermic; however, the temperature rises are not significant and occur locally at the position where such bonds are formed. Even the spherical shape of the microsilica grains, which more easily enter between the free spaces contained between the polymer chains, may be of some importance here. In contrast, this process does not occur for carbon nanotubes, because, as already mentioned, they connect to the polymer by forming two interdigitating networks. Both of them behave similarly with regard to their ability to form new chemical bonds. Of the modified series, the SD/S/N series exhibits the highest polar SFE, and the lowest dispersive component. Despite the seemingly small changes in these values, they can be of great importance in the final formation of adhesion to the substrate.

### 3.2. Adhesion of Glue to Concrete Substrate

The concrete substrate to which the CFRP tape sections were glued was thoroughly cleaned and sandblasted before gluing. After this process, measurements were made with a contact profilometer to determine the height and amplitude parameters. Their results are shown in Figure 13 and Table 4, while a more detailed discussion is described in [30,41]. As expected, the sandblasting process led to significant differentiation of the concrete surface, the detachment of the so-called weak layers, the removal of areas with cement milk and sand grains not permanently bound to the substrate. In addition, it is worth noting the R_ku_ parameter. If its value is ≈3, it means there are elevations with smooth surfaces and steep slopes of depressions.

The effectiveness of bonding is most influenced by three factors: the depth of the cavities R_v_, the width of the grooves R_sm_ and the arithmetic factor of the coordinates of the profile deviations R_a_. The other parameters have an important auxiliary function in evaluating the profile parameters. The influence of mechanical adhesion and the way the adhesive penetrates the irregularities depends most on these values.

As the final stage of the study, the pull-off measurements were performed. Their results are shown in Figure 14.

Among the modified formulations, the series adhered with adhesives with microsilica (66.1% increase) and carbon nanotubes (27.8% increase) showed an increase in adhesion. In contrast, a decrease of 15.7% was recorded for the SD/S series. The differential impact of each of the modified series is clearly seen. In the case of the SD/S series, the adhesive reached a lower final viscosity compared to the unmodified adhesive. This translated into some specific changes in the values of the SFE components, primarily a decrease in the value of the polar component. As a result, it contributed to lower adhesion under the CFRP tape bonding conditions. This is due to the lesser contribution of the adhesion. It is known that cavitation in the SD/S series caused secondary organization of the resin structure, especially the resin binder [30,43]. In this way, the number of electrons, which could bind to the substrate on an exchange basis after the formation of adsorption forces, decreased. The SD/S/M series showed a significant effect of microsilica addition on the final adhesion of the resin to the concrete substrate, similar to other studies [41,60]. As already shown, the adhesive with the addition of this filler showed a marked increase in viscosity, which influenced better adhesion of the adhesive when it was applied to the concrete. Further correlations arise from the way in which the adhesive is placed and penetrates the irregularities of the substrate during the crosslinking and hardening process of the adhesive. Despite the lowest value of the polar component for this series, the highest value was reached by the dispersive component. However, it was lower than the value shown for the SD series. As the results indicate, a kind of compromise between the optimal effect of the microsilica additive on SFE, viscosity and final adhesion was reached. Of great importance in this case is the process of dispersion of microsilica between the complex structure of the adhesive, which consists of individual mers linked in chains, and the quartz powder. The increase in the dispersive component indicates a better “fit” of the adhesive to the topography of the substrate. As a result of the occurrence of strong mechanical adhesion, the adhesive binds better to the substrate through mechanisms of wedging and fitting to the substrate. Of great importance, of course, is the method of preparation itself. In the case of other known methods of preparing the concrete surface (grinding, cleaning), the effect can be quite different. The adsorption also contributes to shaping the growth of SD/S/M adhesion. While the polar component itself clearly decreases, it is possible that higher energy and strength are possessed by temporary bonds between the modified adhesive and the substrate. At this point, adhesion resulting from the theory of acids and bases, the so-called Lewis theory, may be relevant. It assumes the definition of an acid as a molecule or group of molecules capable of accepting electrons, and a base as molecules capable of donating electrons. In the concrete substrate–SD/S/M adhesive system, the acid is a resin containing microsilica, while the base is molecules from the concrete substrate, containing, e.g., metal atoms that make up the cement. Sandblasting did not uncover the granite aggregate grains used to make the concrete, so it is the cement binder that shapes the substrate’s ability to transfer electrons. The exchange of electrons occurs at a very high rate, and the process itself can be initiated only after the adhesive is applied to the concrete.

The increase in the adhesion of the SD/S/N series should be explained somewhat differently. This adhesive, as shown in Figure 9, had the highest viscosity. This is also confirmed by studies on the adhesion of epoxy resins with carbon nanotubes to, for example, carbon and aramid fibers [50,54,57,61]. The polar component was also the highest among the modified series, but the lower value of the dispersion component reduced to some extent the ability of the adhesive structure to match the unevenness of the substrate. Of great importance in this case is the network formed by breaking down the nanotube structure, which in a sense stiffened the structure of the adhesive itself. In this way, the influence of mechanical adhesion was reduced slightly. The carbon nanotube particles bond differently to the epoxy binder, they do not fill the free spaces between the chains as the spherical microsilica particles do but permeate the polymer network. Instead, the presence of free double bonds broken from the original nanotube structure increases the adhesive’s ability to bind to the substrate and form adsorptive bonds. As with the SD/S/M series, it is possible that the effect of using carbon nanotubes on a concrete substrate prepared according to a different method would also be different.

Additionally, as can be seen in Figure 15, the introduced modifications influenced the mode of detaching the pull-off discs from the sandblasted surface. In the case of the modified series, there is a noticeably larger amount of detached substrate. This indicates stronger bonds between the adhesives and the concrete. The effect was achieved to improve both mechanical adhesion (the adhesive fills the surface irregularities more precisely) and adsorptive adhesion (the occurrence of a larger number of bonds). No destruction of the tape itself or detachment of the adhesive without a fragment of the substrate was observed for any of the series. The properties of epoxy resins influence both types of adhesion. The structure of the resin and the particles in polymer chains can easily bond to the substrate. This shows a certain level of reactivity. The chemical bonds formed in this way are of an adsorption nature. The use of modifications gives the resin a certain degree of ability to form such connections. Their intensity may result from the distribution of active adhesion centers on the concrete surface itself; their distribution is random but relatively even. In order for proper bonds to be formed, the adhesive must be properly distributed over the concrete surface. The effectiveness of this treatment largely depends on the dispersion properties. The previously described method of covering any unevenness with the glue is also important. It is worth noting in mind that the physicochemical state of the adhesive particles on its surface is different from in its deeper parts [17,18]. The adhesive molecules on its surface, striving to achieve a state of equilibrium, strive to create a unitary surface as easily as possible from the energy point of view. It is possible to further improve these properties using other fillers, e.g., cement, metakaolin, clay flour.

## 4. Conclusions

The conducted research demonstrated the validity of using sonication as a method of modifying the adhesive, but with the additional use of fillers. The SD/S series showed no increase in adhesion (15.7% decrease). In this case, the use of microsilica, which allowed the adhesion of the adhesive to the sandblasted concrete substrate to be significantly increased (by approximately 66%), was justified. For the carbon nanotubes, the impact on the increase in adhesion was smaller but also significant (approximately 28%). Viscosity, SFE and adhesion of the adhesive to the substrate are reliable indicators for assessing the effects of modification and explaining the phenomena occurring in the adhesive structure under the influence of sonication. However, the changes in these parameters are varied, as shown by the results in Figure 9, Figure 11 and Figure 12. As shown by the example of the SD/S/N series, a significant increase in viscosity (over three times) does not necessarily translate into the same degree of increase in adhesion. This is due to the inability of the adhesive to freely fill the depressions on the concrete surface at such a high viscosity. The addition of microsilica allowed a certain optimal state to be achieved between the filler content (0.5%), an increase in viscosity (by about 63%), a slight reduction in SFE (about 8%) and a final increase in adhesion. It is worth noting that the fillers used in the tests do not affect the course of the resin crosslinking reaction. However, changes in viscosity may affect the necessity of its appropriate use in practical conditions of strengthening reinforced concrete elements with CFRP tapes. The obtained results can serve as a reference point for combining the modified adhesive with other types of substrates, as well as for searching for other fillers whose addition could beneficially change the adhesion. In addition, it is also possible that the fillers used will be able to reduce the oxidation effect of polymers that occurs over longer periods of use. The expected impact of fillers on this process is to retain electrons leaving the polymer structure during oxidation within the network of van der Waals bonds between the filler and polymer molecules. This type of analysis requires a long period of observation and will be carried out as the next stage of research on the effectiveness of the applied modifications.

## Figures and Tables

**Figure 1 materials-16-07408-f001:**
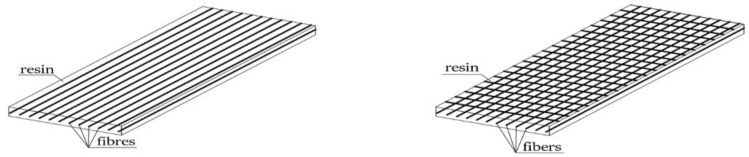
Structure of the FRP composites.

**Figure 2 materials-16-07408-f002:**
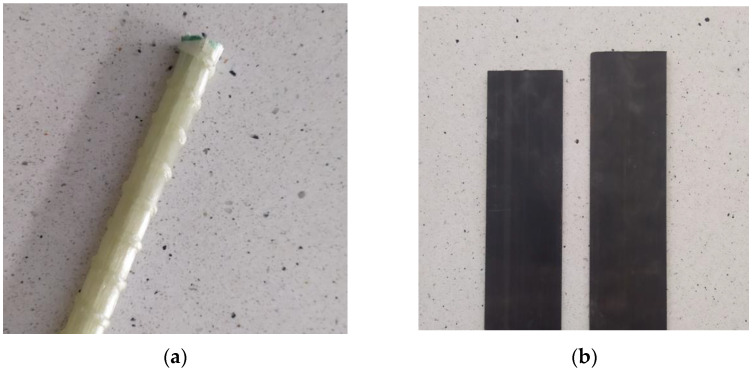
The FRP elements: (**a**) GFRP-reinforced bar, (**b**) CFRP-reinforced tape.

**Figure 3 materials-16-07408-f003:**
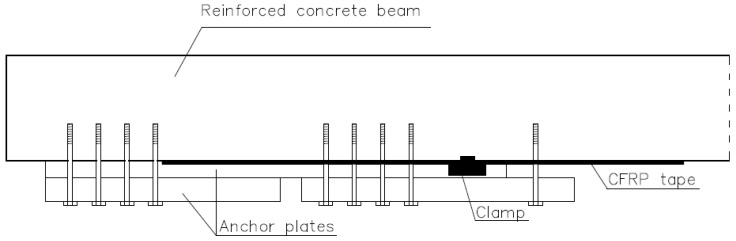
Diagram of active anchoring of the CFRP tape [18,22].

**Figure 4 materials-16-07408-f004:**
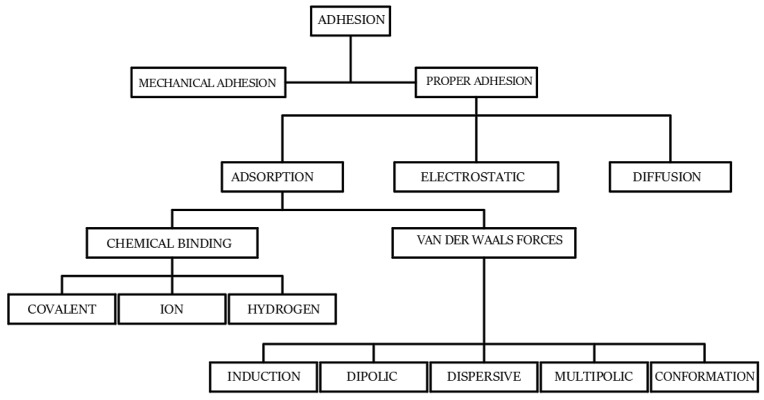
Models of adhesion [19,20,21].

**Figure 5 materials-16-07408-f005:**
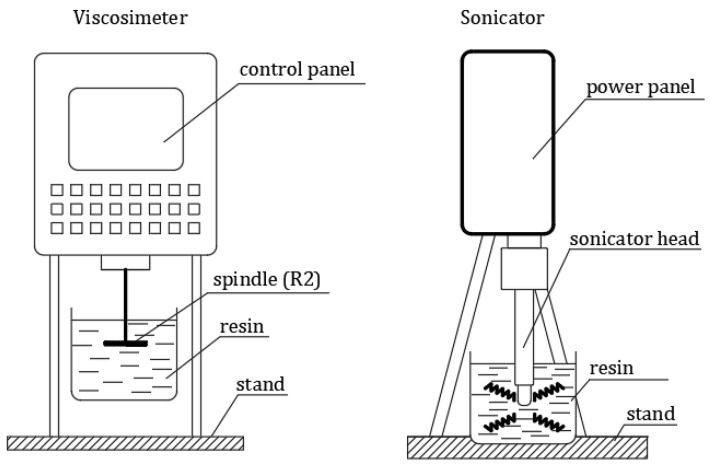
Scheme of the viscosity and sonication test stands.

**Figure 6 materials-16-07408-f006:**
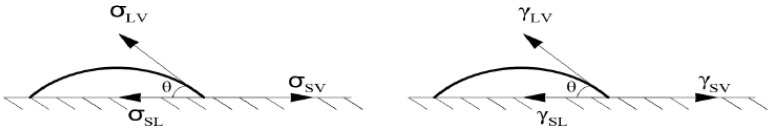
Distribution of surface tension forces and surface energy at the solid–liquid interface. Σ—surface tension; γ—surface free energy; S—solid phase; L—liquid phase; V—gas phase; double indexes refer to the boundary of individual phases.

**Figure 7 materials-16-07408-f007:**
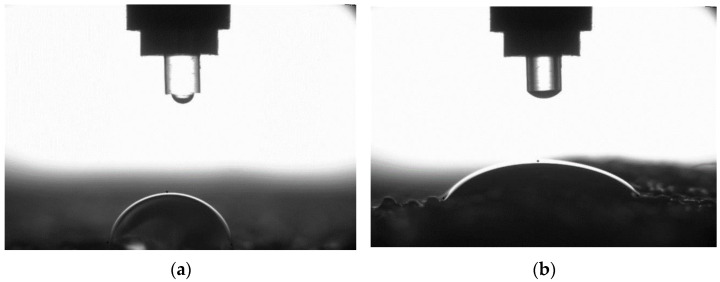
Droplet view of (**a**) distilled water, (**b**) diiodomethane on the surface of an SD/S series sample.

**Figure 8 materials-16-07408-f008:**
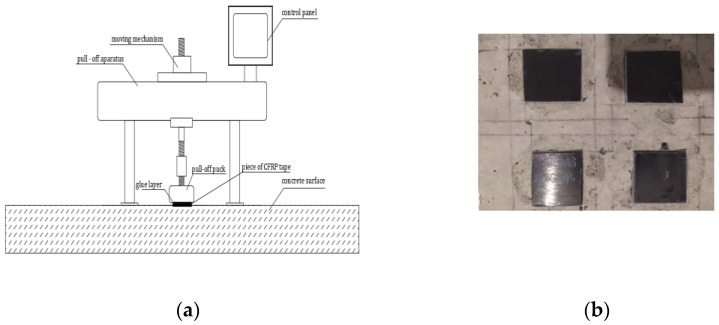
The pull-off research position (**a**) and the CFRP tape samples glued to concrete surface (**b**).

**Figure 9 materials-16-07408-f009:**
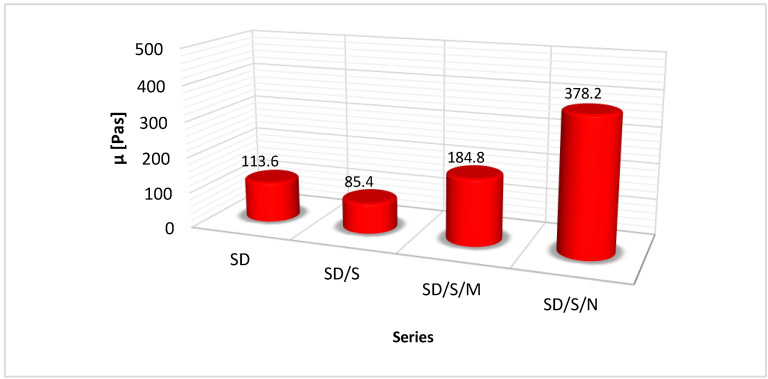
Results of viscosity measurements of the tested resins at 22 °C, µ—viscosity.

**Figure 10 materials-16-07408-f010:**
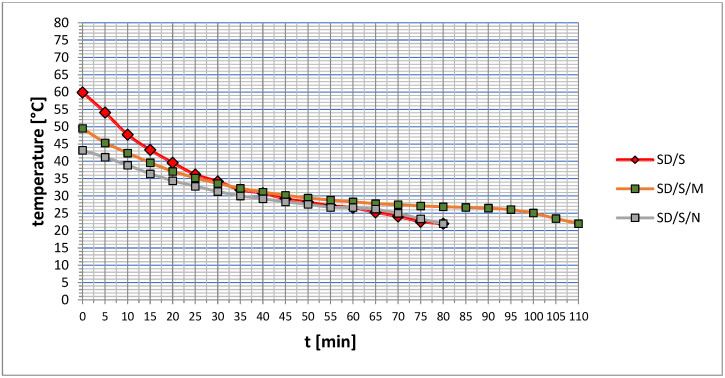
Changes in viscosity and temperature at 5 min intervals.

**Figure 11 materials-16-07408-f011:**
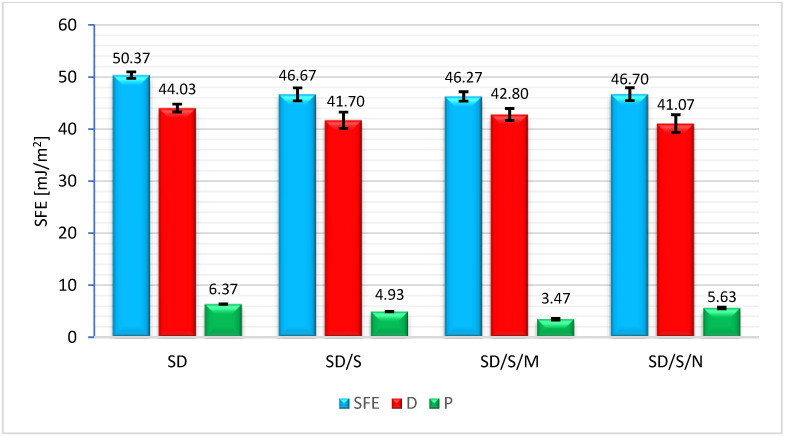
Results of the surface free energy measurements. D—dispersive component; P—polar component.

**Figure 12 materials-16-07408-f012:**
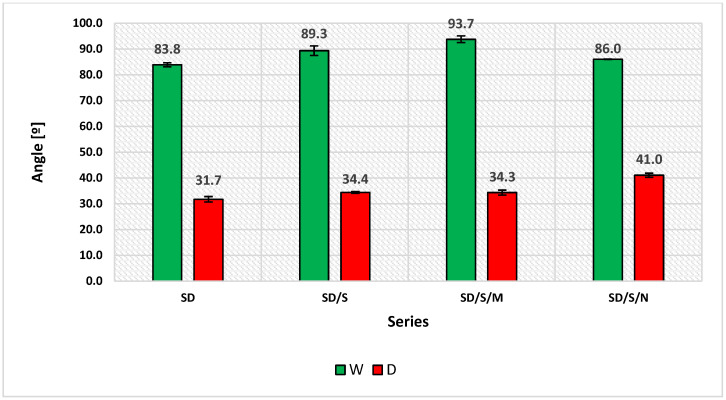
Results of measurements of wetting angles of distilled water (green, W), diiodomethane (red, D).

**Figure 13 materials-16-07408-f013:**
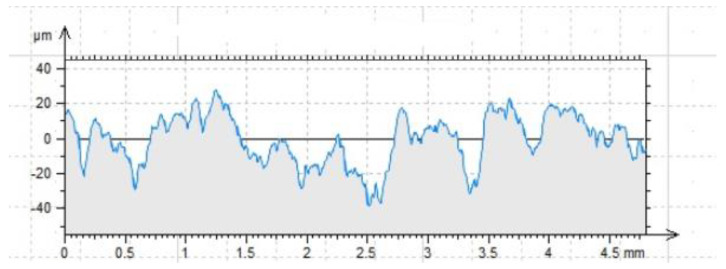
The profilogram of the sandblasted surface. Length = 4.80 mm; Pt = 66.6 μm; scale = 100.00 μm.

**Figure 14 materials-16-07408-f014:**
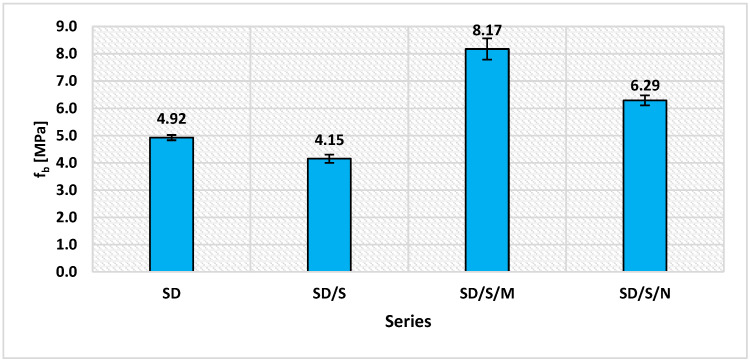
Results of the pull-off tests; f_b_—pull-off adhesion [MPa].

**Figure 15 materials-16-07408-f015:**
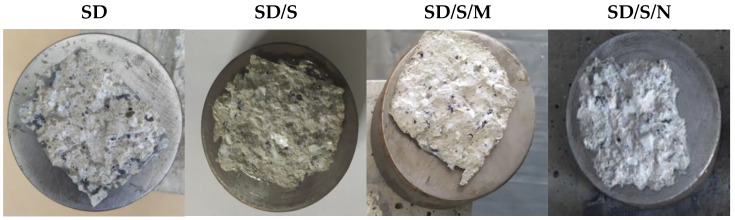
View of the pull-off discs detached from the concrete.

**Table 1 materials-16-07408-t001:** Mechanical properties of FRP composites, * steel for comparison [2,5,6,7].

FRP Composite	Tensile Strength [MPa]	Modulus of Elasticity [GPa]	Density [g/cm^3^]
GFRP	480–4580	35–86	1.25–2.5
CFRP	600–3920	37–784	1.5–2.1
AFRP	1720–3620	41–175	1.25–1.45
BFRP	600–1500	50–65	1.9–2.1
Steel *	280–1900	190–210	7.85

**Table 2 materials-16-07408-t002:** Properties of the epoxy adhesive used in tests.

Kind of Resin	SD
Form	gray, dense mass
Flashpoint [°C]	85
Gelation time [h]	3
Epoxy number [mol/100 g]	0.51–0.55
Density (22 °C) [g/cm^3^]	1.65
Viscosity (22 °C) [Pa·s]	115
Solubility	ketones, esters, alhohols

**Table 3 materials-16-07408-t003:** Recipes and series used in research.

Series	Resin Type	Type of Additive/Modification	Amount of Filler [%]	Amount of Hardener [%]
SD	Epoxy + quartz powder	-	-	33
SD/S		sonication	-	33
SD/S/M		sonication + microsilica	0.5	33
ER52/S/N		sonication + carbon nanotubes	0.1	33

**Table 4 materials-16-07408-t004:** Roughness profile parameters for sandblasted surfaces.

Surface	R_p_	R_v_	R_z_	R_c_	R_t_	R_a_	R_q_	R_sm_	R_sk_	R_ku_
S	15.5	21.1	36.5	22.6	47.4	6.95	8.74	0.3	– 0.467	2.89

## Data Availability

Data available in a publicly accessible repository.

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
