# Peer review of "Changes in the Rheological and Adhesive Properties of Epoxy Resin Used in the Technology of Reinforcement of Structural Elements with CFRP Tapes"

_materials, 2023, doi:10.3390/ma16237408_

Round 1

Reviewer 1 Report

Comments and Suggestions for Authors

This study shows that the use of micro-silica and carbon nanotubes can modify the adhesion characteristics of a resin on a concrete surface.

The paper is well organized, and the characterization is well done.

I recommend this work for publication after the following points have been addressed:

1)     The introduction section should be updated considering the following works:

  https://doi.org/10.1016/j.compositesb.2014.09.022
https://doi.org/10.3390/ma10101131

https://doi.org/10.1016/j.compositesb.2014.01.032

2)     In the section "Materials and Methods" more information should be added about the type of used resin (characteristics and suppliers) and curing procedure (temperature, time, etc.), as described for the individual fillers.

3)     The authors along line 359 state: "The presence of free bonds results in the possibility of permanent and temporary chemical bonds between the nanotubes and the polymer". This statement is somewhat risky if not corroborated by FTIR analyses or relevant literature. The presence of bonds between carbon nanotubes and resin presupposes the presence on the walls of the nanotubes of functional groups that can react with functional groups present in the resin. Since the text does not refer to the structure of both (carbon nanotubes and resin), it is advisable to re-modulate the written sentence by citing possible interactions between CNTs and resin with accompanying literature on the matter.

4)     Authors should check the inclusion of references. Along line 235 of the text, a citation is missing.

5)     The authors should give a brief description, in the text, of the fb parameter shown in Figure 14.

6)     In the conclusions, the authors state: "The conducted studies showed the validity of using sonication as a method of adhesive modification however with the additional participation of fillers." Although the effect of sonication is present in the various characterizations, the main role is attributed to the use of different fillers (see rheological characterization and adhesion tests). The authors should give more emphasis to this effect rather than to the sonication process. It is advisable to modify the written sentence in the conclusions. Furthermore, the conclusions must be implemented with the main quantitative results obtained.

Comments on the Quality of English Language

 Minor editing of English language required

Author Response

With reference to the review of the article entitled "Changes in the rheological and adhesive properties of epoxy resin used in the technology of reinforcement of structural elements with CFRP tapes", a changed version was sent. All changes introduced: redrafting the introduction, supplementing the discussion with analyzes related to the condition of the glue and its adhesive properties, extending the summary with qualitative data, changes in the form of presentation of the materials used, were made in accordance with the reviewers, addition of some articles connected with the main subject of the research' recommendations. I hope their inclusion is done correctly.

Reviewer 2 Report

Comments and Suggestions for Authors

An interesting and important work has been carried out on the modification of epoxy resin to improve the interface bonding performance of cfrp-concrete. The paper is well designed to study the effects of two kinds of fillers and different treatment processes on the rheological and adhesion properties of epoxy resin. The authors are encouraged to consider the following comments to make further responses and improvements. 

1.     Please provide some detailed research results and important findings including some quantitative data analysis in the abstract section. In addition, please provide further information about the effect mechanism of processing technology and fillers on the interface bonding properties.

2.     Why can the modified epoxy resin reduce the performance degradation caused by oxidation? Please provide the relevant explanations.

3.     In terms of main materials of this paper, in addition to excellent mechanical properties, the long-term properties of CFRP under the harsh service environment, such as fatigue properties, creep properties and durability, are also some major advantages that need to be considered in engineering applications compared to the other FRPs. It is suggested to consider the above comments and make corresponding supplements by reviewing the related latest research, such as Composite Structures, 2022. 281: 115060. Polymers, 2023, 15: 2483.

4.     For a research paper, the current summary of introduction work is too long. It is suggested to make a targeted summary of research work closely related to this paper by condensing the current writing. In addition, please further explain in the introduction why you want to use silicon dioxide and carbon nanotubes as the filler of epoxy resin.

5.     In the section of methods, it is suggested to add Arabic numerals (1), (2), (3)… to each method and add the titles of related test methods.

6.     Please explain the basis and source of addition content of two fillers in this paper in the material section. Can this addition content ensure that the epoxy resin has the best adhesion properties?

7.     How to evaluate the influence of viscosity on the process performance and interfacial bonding performance of epoxy resin? For epoxy resin as adhesive, what is the optimal value of viscosity when considering the above two factors?

8.     What is the synergistic mechanism of two on the surface properties and bonding properties of epoxy resin? It is suggested to add relevant analysis and discussion.

9.     In Section 3.2, please provide the interface failure mode after pull-off tests and the improvement mechanism after adding the fillers.

10.  Please provide some important quantitative result data in the conclusion section. In addition, the bonding performance improvement mechanism should also be reflected in this part.

Author Response

(The authors gave the same response as above.)

Round 2

Reviewer 1 Report

Comments and Suggestions for Authors

The reviewer's comments have been sufficiently addressed. Therefore, the manuscript can be accepted for publication.

Comments on the Quality of English Language

Minor editing of English language required

Reviewer 2 Report

Comments and Suggestions for Authors

The paper can be published after revision.